# The Relationship between VO_2_max, Power Management, and Increased Running Speed: Towards Gait Pattern Recognition through Clustering Analysis

**DOI:** 10.3390/s21072422

**Published:** 2021-04-01

**Authors:** Juan Pardo Albiach, Melanie Mir-Jimenez, Vanessa Hueso Moreno, Iván Nácher Moltó, Javier Martínez-Gramage

**Affiliations:** 1Embedded Systems and Artificial Intelligence Group, Universidad Cardenal Herrera-CEU, CEU Universities, 46115 Valencia, Spain; melanie.mir@alumnos.uchceu.es; 2Department of Physiotherapy, Universidad Cardenal Herrera-CEU, CEU Universities, 46115 Valencia, Spain; ivan.nacher.molto@gmail.com (I.N.M.); jmg@uchceu.es (J.M.-G.); 3Triathlon Technification Program, Valencian Community Triathlon Federation, 46940 Manises, Spain; vanessa.huesa@triatlocv.org

**Keywords:** VO_2_max, power, running biomechanics, hierarchical cluster analysis, machine learning, triathletes

## Abstract

Triathlon has become increasingly popular in recent years. In this discipline, maximum oxygen consumption (VO_2_max) is considered the gold standard for determining competition cardiovascular capacity. However, the emergence of wearable sensors (as Stryd) has drastically changed training and races, allowing for the more precise evaluation of athletes and study of many more potential determining variables. Thus, in order to discover factors associated with improved running efficiency, we studied which variables are correlated with increased speed. We then developed a methodology to identify associated running patterns that could allow each individual athlete to improve their performance. To achieve this, we developed a correlation matrix, implemented regression models, and created a heat map using hierarchical cluster analysis. This highlighted relationships between running patterns in groups of young triathlon athletes and several different variables. Among the most important conclusions, we found that high VO_2_max did not seem to be significantly correlated with faster speed. However, faster individuals did have higher power per kg, horizontal power, stride length, and running effectiveness, and lower ground contact time and form power ratio. VO_2_max appeared to strongly correlate with power per kg and this seemed to indicate that to run faster, athletes must also correctly manage their power.

## 1. Introduction

Triathlon is an increasingly popular sport with broad participation spanning three disciplines (swimming, cycling, and running) in the same event. In recent years in Spain, participation in triathlon has increased by more than 200% among young athletes of school age (Spanish Triathlon Federation) [1]. In this discipline, maximum oxygen consumption (VO_2_max) is considered the gold standard for determining cardiovascular capacity [2]. Accurate VO_2_max measurement requires specialised equipment found in exercise physiology laboratories—techniques that are often not available to every professional. In addition, testing an entire team can be time consuming because only one athlete can be evaluated at a time. Therefore, alternative parameters have been developed to predict VO_2_max that allow several athletes to be tested at the same time without requiring sophisticated laboratory tools [3].

The ability both to maintain a high percentage of VO_2_max for long periods of time and simultaneously move efficiently, referred to as running effectiveness (RE), comes from a series of physiological attributes that contribute to the success of running performance and help athletes stand out [4]. RE is generally used to refer to steady-state oxygen consumption at a given running speed and expresses the energy expenditure required by individuals to perform at a given intensity [5]. Trained runners have higher REs compared to lesser-trained runners, which indicates that positive adaptations occur in response to regular training. Although a given athlete may be genetically predisposed to having a ‘good’ RE, various strategies can potentially further enhance an individual’s RE by increasing metabolic, cardiorespiratory, biomechanical, and/or neuromuscular responses [4].

Until a few years ago, RE was not considered an important factor in the improvement of athletes’ careers. However, this area is now the focus of increasing interest. RE is the result of the interaction between multiple factors. Of these, the most important may be biomechanical factors, neuromuscular variables such as leg stiffness, exposure to training periods at altitude, and anthropometric variables [5]. A good correlation has been observed between RE and oxygen consumption (VO_2_) while running. Runners with a good RE use less oxygen than runners with a poor RE at the same speed and under homogeneous conditions [6]. However, it has also been noted that RE can vary by up to 30% between trained runners with a similar VO_2_max [7].

In recent years, the advent of portable power estimators has dramatically changed training and competitive running, allowing athletes to be accurately evaluated [8]. Among these systems, Stryd, Boulder, CO, USA (www.stryd.com, accessed on 1 March 2021) pioneered the manufacture of power meters for runners. The Stryd running power meter is a pedometer that attaches to the shoe to measure variables that quantify performance including pace, distance, elevation, power, form power, cadence, ground contact time, vertical oscillation, and leg spring stiffness [9].

This is a relatively new type of instrument, and the validity and reliability of these systems for evaluating power output and space–time parameters have only recently been validated. In this context, the operating power data recorded by Stryd has been successfully used to establish a linear relationship between power and speed to predict power output at different submaximal operating speeds, demonstrating the great potential of this portable equipment for studying efficiency patterns while running. Additionally, a few studies found a positive correlation between Stryd’s power data and the operating economy or metabolic demands. Indeed, a recent study by Cerezuela-Espejo et al. determined the correlation between these power meters and oxygen consumption [8]. Moreover, Cartón-Llorente et al. 2021 [10], determined that Stryd could reliably determine the functional threshold power (FTP) of runners.

The detection of running patterns and the variables involved in achieving the maximum possible speed while running has always been the subject of research [11,12,13]. This allows us to compare which parameters best define running efficiency, meaning that the similarities and discrepancies between athletes who are more or less successful in competitions can be examined. In this sense, the use of objective grouping or classification techniques (which are commonly employed with a variety of goals in different fields such as engineering, science, or technology) is also feasible in sports sciences. Thus, unsupervised classification (commonly known as clustering) is a classical technique used in the area of machine learning [14]. According to Rokach [15], clustering divides data patterns into subsets in such a way that similar patterns are grouped together.

Several studies have focused on gait patterns by using clustering techniques such as hierarchical clustering analysis (HCA). These provide an interpretable analysis of large quantities of data from sensors, as a multivariate problem, to obtain different groups of athletes with similar running gait patterns [16]. The objective of this study was to determine running patterns and variables involved in attaining maximum running speed in young triathletes.

## 2. Materials and Methods

### 2.1. Participants

The participants belonged to the high-performance Triathlon Technification Plan based in the Valencian Community in Spain. The study was approved by the Ethics Committee for Biomedical research at the CEU-Cardenal Herrera University, (reference No: CEI18/137) and was registered as a clinical trial (ClinicalTrials.gov registration No: NCT04221698).

#### Inclusion/Exclusion Criteria

Fifteen healthy triathletes (9 males and 6 females) were enrolled in this study (Table 1):

Participants were included if they reported having run a minimum of 2 days per week in the 3 months prior with no reported injuries and with their worst pain rated a minimum of 3 out 10 on a numerical rating scale (NRS) for pain (0 = no pain; 10 = worst possible pain) [17]. Participants were excluded if they reported any previous musculoskeletal surgery, neurological impairment, knee structural deformities, pain caused by trauma or sports activities, having stopped running, or having received additional treatment outside of this study.

### 2.2. Data Collection

All the participants performed a 5 min warm-up on a treadmill (HP Cosmos Quasar, Nussdorf-Traunstein, Germany) at their preferred speed [17]. The initial running speed was set at 8 km/h with a gradient of 1% [18]. The starting speed was 3 km/h, with speed increments of 1 km/h every 60 s. The subjects walked the first three steps (up to 7 km/h), and continued running from 8 km/h, until volitional exhaustion. After exhaustion, the athletes underwent a 5-min recovery protocol during which the speed was decreased each minute from 100% to 60%, 55%, 50%, 45%, and 40% of the maximal achieved speed [19].

Expired gas was sampled continuously and O_2_ and CO_2_ concentration in expired gas were determined using the Ultima™ CardiO2^®^ gas exchange analysis system ((MGC Diagnostics Corporation, St Paul, MN, USA, https://mgcdiagnostics.com, accessed on 1 March 2021). Heart rate (HR) was collected using a telemetric heart rate monitor (Polar Electro, Kempele, Finland), and stored in PC memory. The thresholds assessed were Aerobic and Anaerobic Ventilatory Thresholds (VT1 and VT2), identified by different ventilatory criteria, such as: VSlope (VO_2_ and VCO_2_), Ventilatory Equivalents (EqO_2_ and EqCO_2_), Ventilation (VE), Pressures at the end of each expiration (Pet O_2_ and Pet CO_2_), and Respiratory Quotient (RER).

The Stryd sensor, paired with a Garmin Forerunner 935 watch, was used to determine running power and recording was started and stopped at the same time as the stress test. As shown in Figure 1, the powers at each threshold were recorded at the same time the ventilation thresholds 1 and 2 (VT1 and VT2) and maximal aerobic power (MAP) occurred, with these physiological variables also being defined.

Once the participant data were acquired, a raw data set was constructed for the purpose of this study. We then assessed and cleaned the database to correct possible errors in the data, e.g., missing values or extreme values gathered from the overall system. Then the data was arranged in “csv” format to be treated by the statistical program RStudio [20]. Finally, the experimental data set was structured with 14 columns referring to the measured variables for each participant and 15 rows corresponding to each athlete.

#### Variables Analysed

To determine athlete running power, we used a Stryd sensor (Stryd power meter; Stryd, Inc., Boulder, CO, USA, https://www.stryd.com, accessed on 1 March 2021) a relatively new device, which estimates power in watts. Stryd is a carbon fibre-reinforced foot pod that attaches to the shoe and weighs 9.1 g. The sensor is based on a 6-axis inertial motion sensor (3-axis gyroscope and 3-axis accelerometer). We analysed the following variables: power (W), leg spring stiffness (LSS), leg spring stiffness per kg (LSS/kg), vertical oscillation (VO), power per kg (W/kg), horizontal power (HW), speed (SPD), cadence (CAD), ground contact time (GCT), vertical ratio (VR), stride length (SL), running effectiveness (RE), form power ratio (FPR), and maximum oxygen consumption (VO_2_max). To determine the VO_2_max, the Ultima™ CardiO2^®^ gas exchange analysis system (MGC Diagnostics Corporation, St Paul, MN, USA was used.

### 2.3. Data Analysis

We used different statistical and artificial intelligence data analysis techniques to examine the data we collected. Our objective was to understand which variables most influence running efficiency. Thus, we studied which factors were related to each other based on their linear correlations and tried to understand how some characteristics influence others with the goal of obtaining clues that could explain different running patterns in young triathletes.

Clustering techniques were used to obtain running patterns that would allow us to visually generate groups of individuals with similar running characteristics [14]. These groups were formed based on the data collected and extracted from the Stryd sensor. Thus, each runner had their own colour pattern which we could use to identify the variables each individual should work on to improve both their speed and efficiency. We performed all of the calculations with RStudio desktop software for macOS (version 1.3.1073, ‘Giant Goldenrod’ release) [20]. In the following section, we detailed the techniques we used to understand the interpretation of the results.

#### 2.3.1. Linear Correlations

Linear correlation and simple linear regression are statistical methods that study the linear relationship between two variables. Correlation quantifies how related two variables are, while linear regression consists of generating an equation (model) that, based on the relationship between two variables, allows the value of one to be predicted based on the other. Thus, variables X and Y are said to be positively correlated if high values of X are associated with high values of Y, and low values of X are associated with low values of Y. In contrast, if high values of X are associated with low values of Y, and vice versa, the variables are negatively correlated [21]. Correlation coefficients range from −1 (for a negative correlation) to +1 (for a positive correlation)—correlations close to 0 indicate the absence of a linear correlation between two variables [22].

As a general rule, linear correlation studies precede the generation of linear regression models, after the confirmation of a correlation between variables. The difference is that while correlation measures the strength of an association between two variables, regression quantifies the nature of the relationship [21]. Therefore, it is useful to calculate a correlation matrix showing all the variables in rows and columns, in which the intersection values quantify the correlation between them. This matrix can then be used to calculate a ‘correlation map’ that highlights which variables were linearly related to each other at a statistical given significance level (*p*-value), in our case, *p* = 0.05.

Thus, we were able to quickly render a colour map that quantified the significance and direction of the relationship between two variables, therefore enabling us to choose which ones merited further study. We mainly focused on the speed variable in this current work, although we did study other possible correlations that (through other measurements) could help explain what influences running efficiency. The correlation map also gave us a much better understanding of running patterns.

#### 2.3.2. Hierarchical Clustering Analysis

The term clustering refers to a wide range of unsupervised techniques from machine learning fields whose purpose is to find patterns or groups of similar objects (known as clusters) within a set of observations [23]. Clustering is one of the most important data mining methods for discovering patterns in multidimensional data. The partitions are established such that observations within the same group are similar to each other and different from the observations of other groups. Thus, unsupervised learning can be viewed as an extension of exploratory data analysis to gain insights into a set of data and how the different variables relate to each other. Additionally, clustering provides tools to analyse these variables and discover relationships and patterns within them [23].

An excellent review of clustering techniques can be found in [14] which also describes a common clustering technique taxonomy proposed by Fraley and Raftery [24]. They suggested dividing these techniques into two different groups: hierarchical and partitioning methods. After testing different techniques in this work, we focused on HCA. Although other techniques with different advantages and disadvantages are available, we considered this technique to be best suited to our data set.

HCA is an alternative to the common K-means technique and is more flexible and better able to discover outlying groups or records. This type of clustering also lends itself to intuitive graphical display, leading to easier cluster interpretation. HCA methods form clusters by iteratively dividing patterns using a top–down or bottom–up approach. Hierarchical clustering methods may be agglomerative or divisive. The former follows the bottom–up approach to build clusters starting with a single object and then merging these atomic clusters groups of increasing size until all of the objects are finally lying in a single cluster or certain termination conditions are satisfied. The latter is a top–down approach which breaks a cluster containing all objects into smaller groups, until each object forms an independent cluster or the termination conditions are satisfied. The hierarchical methods usually lead to the formation of dendrograms that allow the resulting groupings to be visualised.

## 3. Results

First, we studied the reliability of the Stryd sensor against the gold standard measured in the laboratory by calculating the correlation of the values obtained with the sensor and the standard at each of the three thresholds. Our data demonstrated the reliability of the Stryd compared to laboratory systems, as also shown in some recent studies [8,25] and so we used this data in the subsequent detection of running patterns. Thus, as shown in Table 2, we compared the speed obtained in the laboratory system with the values for W, W/kg, HW, and FPR obtained by the Stryd device. In addition, these data were also compared with VO_2_ (mL/kg/min) measured in the laboratory so we could find the variables that best correlated with speed.

The strongest correlations with speed at each threshold were W/kg, HW, and FPR; on the contrary VO_2_max was not significantly correlated with speed at any of the thresholds.

Figure 2 shows the graphs corresponding to the regression models calculated to compare W/kg and VO_2_max for each of the three thresholds. This allowed us to identify which variable best explained the dependent variable of speed. In this case, the regression models highlighted two variables as explanatory factors for the speed reached by the study participants.

Power, but not VO_2_max, perfectly explained speed for each of the thresholds. As shown in Table 2, there was an exceptionally strong (near 100%) correlation between power and speed, which was also observed in the regression model with an *R*^2^ remarkably close to 1. This indicated that variability in speed could be explained very well by the power of the athlete. Moreover, the regression model indicated how much power would be required to acquire a determined speed at each threshold. On the contrary, this effect was not observed for VO_2_max, and the corresponding regression models could not explain the increase or decrease in speed based on this parameter. There was no evidence to indicate a linear relationship between VO_2_max and speed.

### 3.1. Correlations Map

We carried out both Pearson’s r analysis (the most commonly used method to assess correlations) and Kendall and Spearman correlations as non-parametric methods commonly used to perform rank-based correlation analysis [23]. Nevertheless, the significant correlations remained the same in both cases and so we used Pearson’s correlation coefficient (r) to calculate the correlation matrix to highlight the most pertinent variables to study. As shown in Figure 3. The dots in red tones referred to negative correlations. For example, as the HW variable increased, the FPR decreased, in a significant and quite strong linear correlation remarkably close to −1. In addition, as FPR decreased, GCT decreased, and SPD, W/kg, HW, W, and LSS increased. Furthermore, as GCT decreased, LSS/kg, VO_2_max, SPD, W/kg, and HW increased. Finally, as RE increased, SPD and SL also increased.

Dots in blue tones referred to positive correlations such as the reasonable correlation between SPD and W/kg and significant correlation between SPD and HW. This means that the more W/kg and HW, the higher the SPD runners attained—a clear indicator of running pattern. When LSS/kg increased, CAD also increased and the increase in VO_2_max correlated with the increase in W/kg. As SPD increased, W/kg and HW also increased; increased W/kg produced increased HW, VO, W, and LSS; as HW increased, W and LSS increased; and increased W resulted in increased LSS.

In contrast, some variables were not significantly correlated and when we cross-referenced these there was only one gap in the matrix. For example, our data indicated that a higher running cadence did not mean that the athlete would run faster. Indeed, this variable did not show a significant linear correlation, meaning that, a priori, it was unlikely to be an important factor in the generation of more speed. In contrast, there was no correlation between athletes with a high VO_2_max and faster speed, as we previously observed in our regression models. However, faster athletes had a higher W/kg and HW, and a lower FPR. VO_2_max strongly correlated with W/kg and this seemed to indicate that to run fast, athletes must also correctly manage their power.

### 3.2. Clustering Heat Map

Finally, we decided to study the patterns of each runner by generating a heat map using HCA. First the data was scaled to standardise the variables and minimise the impact of the different magnitudes. Thus, the data were normalised to have zero mean and unit variance. When the data were scaled, the Euclidean distance of the z-scores was the same as the correlation distance. On the other hand, a connectivity-based clustering or HCA approach was used to identify homogeneous gait patterns in the entire participant group by creating a cluster tree or dendrogram. To perform the HCA, we used the R package ‘pheatmap’ library (Version 1.0.12) [26]. This allowed us to generate clusters of similar runners based on the variables extracted from the Stryd data and to construct a heat map to observe these patterns according to assigned colours.

The procedure for performing agglomerative HCA on the data set consisted of three steps: calculation of the distance matrix between participants, computation of a linkage function, and definition of clusters. In brief, first the Euclidean distance between every pair of athletes was calculated for an M-dimensional space. Second, individual participants were paired into binary clusters based on the distance information using the Ward D2 linkage method [27]. Third, newly formed clusters were grouped into larger clusters until the dendrogram was formed [16,24]. The Ward minimum variance method was used to minimise the total within-cluster variance. At each step, the cluster pair with a minimum between-cluster distance was merged.

Finally, we visually inspected the dendrogram and decided to separate the clusters into three groups based on our knowledge of the athletes. Thus, the K parameter was established at 3. As shown in Figure 4, we represented the result of the clustering as a heat map.

As shown on the heat map, three clusters of athletes with similar characteristics to each other were identified. The reference group was cluster two (athletes S7 and S13), representing the two individuals with the best competitive results. As shown, the SPD variable for these two athletes corresponded to the highest values, highlighted in warmer colours (red tones). In contrast, the participants with SPD marked in cooler colours (blue tones) were the slowest from among the cohort. The colour scale was established by columns, with red being representing the individual with the highest value in each of the variables.

Thus, a colour pattern could be observed for each individual with respect to the reference group by noting the variables for which warmer or cooler colours were obtained. For example, participant S14 obtained low SPD, VO_2_max consumption, HW, and W/kg values and high values for FPR and GCT, indicating the aspects of their running technique they should work on to increase their RE or running speed. In contrast, athlete S2 had high W/kg, LSS, W, and VO values and low CAD values with respect to the reference participants, even though their running speed was normal. This was probably because of the strong correlation between SPD and W/kg and weak correlation between SPD and the other variables.

Based on these data, we carried out a detailed analysis of which characteristics in each athlete were increased or reduced compared to those who had obtained better results. Moreover, by examining certain reference variables such as RE, we observed differences between the participants. Figure 5 shows a graphical representation of the relationship between speed and the variables that best correlated with it, also separating the individuals by each of the cluster groups. These graphs allowed us to better understand the differences between athletes who run faster and who better manage their performance power compared to those who run slower, according to these groups. Group two was used as the reference and was shown in green.

Thus, the fastest runners had a decreased FPR (A) and GCT (B), and an increased W/kg (C), HW (D), SL (E), and RE (F). Based on these results, it appears that power management and running dynamics play a more important role than VO_2_max in athletes who run faster.

## 4. Discussion

The objective of this study was to determine the running patterns and variables involved in the maximum running speed of young triathletes. We observed that there was a pattern of decreased FPR and GCT, and increased W/kg, HW, SL, and RE among faster athletes. Based on these results, it appears that power management and running dynamics play a more important role than VO_2_max in athletes who run faster. Various studies have demonstrated the reliability and validity of portable systems such as Stryd for measuring running power [9,28,29]. Additionally, running power is a more sensitive measure of exercise intensity than other internal and external parameters, such as heart rate or speed [28].

Calculation of the linear correlations for each of the variables we collected in this study was an easy and fast method to understand which factors or characteristics were related to each other. This allowed us to quickly find indications about the influence of some of these variables with respect to others in order to obtain the most important power parameters for running. We observed that the Stryd device data correlated well with VO_2_max laboratory equipment data. This added confidence to our study of the interrelation of variables and subsequently, to our comparisons between athletes to reliably apply grouping techniques to search for patterns representative of RE. It was also interesting to see that certain variables did not linearly influence RE and so, could be discarded for the purposes of this work, or studied using other distribution models.

In addition, we consider the clustering techniques represented by heat maps to constitute an especially useful tool for quickly explaining the differences between different runners. The colour codes allowed us to find similar patterns for each variable collected during the test, which corresponded to the patterns of each competitor. The ability to group athletes by these colour patterns represents RE patterns either based on the reference of athletes who obtained the best competitive results or simply on a pre-determined set of variables. This will allow us to extrapolate these findings to techniques for other sports in which different characteristics are measured.

### Limitations of the Study and Future Activities

One of the limitations of this work was its sample size because it was only sufficient to allow us to obtain preliminary results related to our research topic. However, this work is encouraging and we believe that future work in this area seems very promising. We must also consider that obtaining data for high-performance athletes is quite difficult because they are a very small population and therefore the sample will never be large. Nevertheless, although our sample cohort was small and homogeneous, we would need a larger number of subjects to have sufficient strength of these results to be able to generalise them with confidence to other athletes with similar characteristics. Additionally, for future research, the variable sex should be considered, as it could be a confusing factor when studying the RE.

Finally, the sample size made it difficult to fully utilise the potential of the some of the artificial intelligence techniques available to us. Future work should be directed towards the application of these results in the training of young triathletes to help improve their performance and to determine biomechanical running patterns that complement the present power study using the Stryd sensor in young athletes.

## 5. Conclusions

In this work, we studied how to identify running patterns among young athletes based on data from wearable sensors (such as Stryd) as compared to laboratory equipment results. Our findings indicate that power management was key to maximising running speed. VO_2_max strongly correlated with W/kg, indicating that to run faster, athletes must also correctly manage their power. We used different techniques to identify the relationship between strength and some of the other variables in our data set. Thus, we were able to establish which parameters each athlete should work on to enhance their running form. Heat maps were a tool that also allowed us to quickly group runners with similar characteristics, defining colour patterns to characterise them. Furthermore, by comparing each athlete’s performance with the other competitors, we were able to work with individual runners to set target parameters for their improvement. Given that the data was obtained from measurement sensors, we consider it to be very valuable and totally objective information that could perhaps lead to the modification of certain methodologies or training techniques. This work opens the door for future work with other types of variables, such as biomechanics obtained from other sensors, which will broaden the spectrum of factors that can be studied.

## Figures and Tables

**Figure 1 sensors-21-02422-f001:**
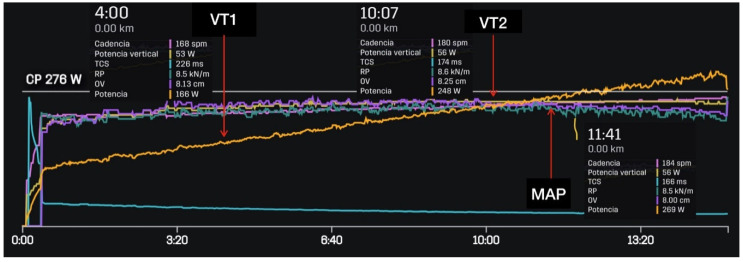
Determination of the ventilation thresholds 1 and 2 (VT1 and VT2) and maximal aerobic power (MAP) with the corresponding power at each physiological threshold.

**Figure 2 sensors-21-02422-f002:**
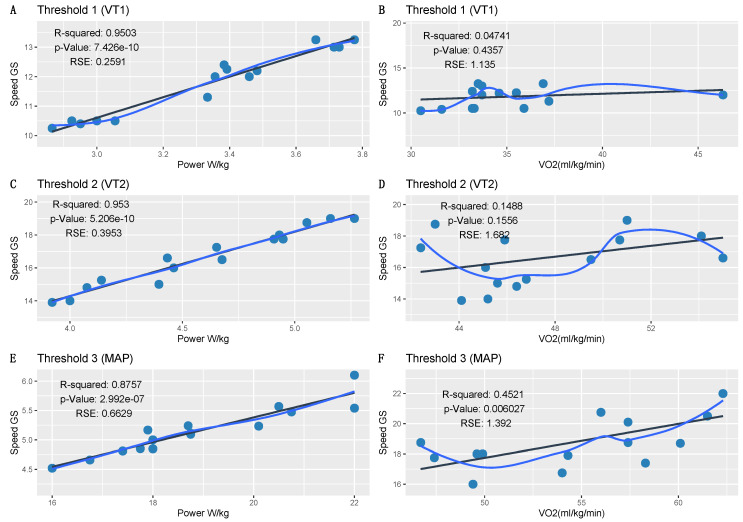
Regression models for power (W) and maximum oxygen consumption (VO_2_max) with respect to speed. The regression models compared power and VO_2_max with speed at ventilation threshold 1 (VT1; **A**,**B**), ventilation threshold 2 (VT2; **C**,**D**), and at the maximal aerobic power (MAP) threshold (**E**,**F**).

**Figure 3 sensors-21-02422-f003:**
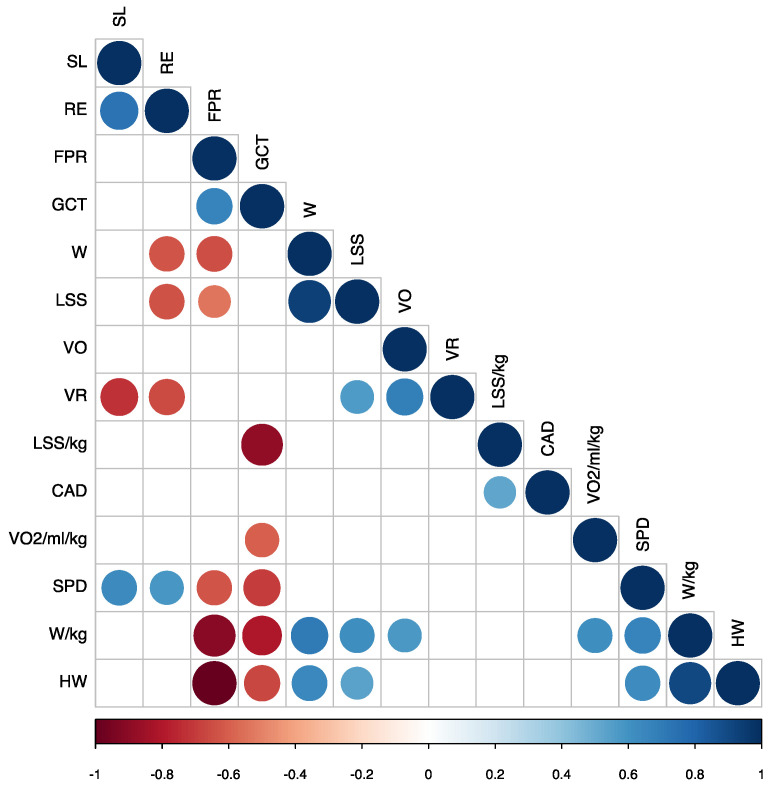
Correlations map representing only the significant (*p* < 0.05) variables. Power (W), leg spring stiffness (LSS), vertical oscillation (VO), power per kilogram (W/kg), speed (SPD), cadence (CAD), ground contact time (GCT), and the form power ratio (FPR).

**Figure 4 sensors-21-02422-f004:**
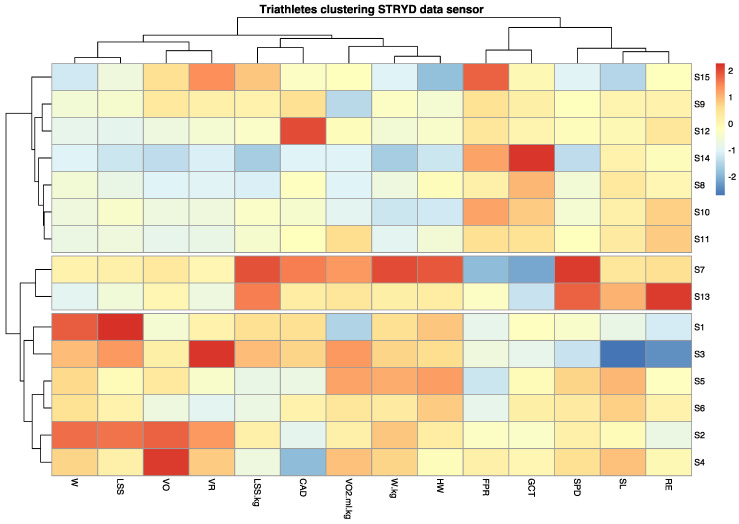
Heat map of the clustering of athletes. The participants distributed into cluster 1 (S1, S2, S3, S4, S5, and S6), cluster 2 (S7 and S13), and cluster 3 (S8, S9, S10, S11, S12, S14, and S15).

**Figure 5 sensors-21-02422-f005:**
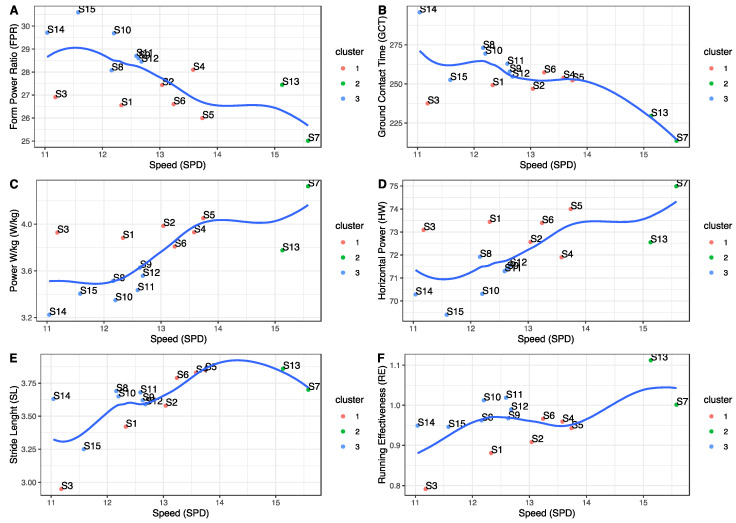
The relationship between speed and the variables that best correlated with it. (**A**) Form power ratio (FPR), (**B**) ground contact time (GCT), (**C**) power per kilogram (W/kg), (**D**) horizontal power (HW), (**E**) stride length (SL), and (**F**) running effectiveness (RE).

**Table 1 sensors-21-02422-t001:** Participant characteristics ^a^.

	Male (*n* = 9)	Female (*n* = 6)
Age	15 ± 1.5	14 ± 1.0
Weight, kg	56.3 ± 8.9	55.2 ± 3.2
Height, cm	170 ± 7.2	168.5 ± 4.3
Body mass index, kg/m^2^	19.4 ± 1.7	19.3 ± 1.2
Years competing	7.8 ± 6.8	6.8 ± 1.0
Training hours per week	19.1 ± 2.8	19.6 ± 2.6

^a^ Values are presented as the mean ± *SD*.

**Table 2 sensors-21-02422-t002:** Correlation between power per kilogram (W/kg), horizontal power (HW), and the form power ratio (FPR) with athlete speed at each running threshold, as well as between VO_2_max and the velocity at each threshold. Note: (ST) refers to the measurement made with the Stryd system.

Threshold	Gold Standard	Variables	Pearson’s Coefficient (r)
VT1	Speed (km/h)	W/kg (ST)	0.97
	Speed (km/h)	HW (ST)	0.82
	Speed (km/h)	VO_2_ (mL/kg/min)	0.22
	Speed (km/h)	FPR (ST)	−0.82
VT2	Speed (km/h)	W/kg (ST)	0.98
	Speed (km/h)	HW (ST)	0.92
	Speed (km/h)	VO_2_ (mL/kg/min)	0.38
	Speed (km/h)	FPR (ST)	−0.92
MAP	Speed (km/h)	W/kg (ST)	0.94
	Speed (km/h)	HW (ST)	0.91
	Speed (km/h)	VO_2_ (mL/kg/min)	0.67
	Speed (km/h)	FPR (ST)	−0.91

## Data Availability

The data presented in this study are available on request from the corresponding author. The data are not publicly available due to privacy restrictions.

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
