# Peer review of "The Relationship between VO2max, Power Management, and Increased Running Speed: Towards Gait Pattern Recognition through Clustering Analysis"

_sensors, 2021, doi:10.3390/s21072422_

Round 1
Reviewer 1 Report
Please see the attached scan of the paper with my embedded comments

Author Response
Response to Reviewer 1 Comments (uploaded a pdf with responses)
Thank you very much for your valuable comments and your time to revise our work. We hope to clarify all your questions.

Reviewer 2 Report
Dear Authors,
you conducted an original and correct research, you presented the results nicely in a very interesting way.
I have just one minor suggestion: I would like to point out a sentence that talks about generalization of conclusions:
-p 371-373 ''Nevertheless, although our sample cohort was small, it was relatively homogeneous. Thus, it is likely that our research conclusions could be generalised to other athletes whose characteristics will be very similar to our sample.''
In order to be able to generalize you would definitely need to have a larger number of respondents. You have to be careful in this. At this sample you don't have the strength to do that. I suggest kindly to you define it differently, and refer to further research that could substantiate this. Remember that you have both sexes in a sample and that this may be reflected in their RE too.
Author Response
Response to Reviewer 2 Comments (uploaded a pdf with responses)
Thank you very much for your valuable comments and your time to revise our work. We hope to clarify all your questions.
